

# Analysis of spatial and temporal patterns of on-road NO$_2$ concentrations in Hong Kong

Ying Zhu[a], Ka Lok Chan[a,b], Yun Fat Lam[c,d], Martin Horbanski[e], Denis Pöhler[e], Johannes Boll[a], Ivo Lipkowitsch[a], Sheng Ye[a], and Mark Wenig[a]

[a]Meteorological Institute, Ludwig-Maximilians-Universität München, Munich, Germany
[b]Remote Sensing Technology Institute, German Aerospace Center (DLR), Oberpfaffenhofen, Germany
[c]School of Energy and Environment, City University of Hong Kong, Hong Kong
[d]Guy Carpenter Climate Change Centre, City University of Hong Kong, Hong Kong
[e]Institute of Environmental Physics, Ruprecht-Karls-Universität Heidelberg, Heidelberg, Germany

**Correspondence:** K.L. Chan (lok.chan@lmu.de)

**Abstract.** In this paper we present an investigation of the spatial and temporal variability of street level concentrations of NO$_2$ in Hong Kong as an example for a densely populated megacity with heavy traffic. For the study we use a combination of open path remote sensing and in-situ measurement techniques that allows us to separate temporal changes and spatial patterns and analyse them separately. Two measurement campaigns have been conducted, one in December 2010 and one in March 2017. Each campaign lasted for a week which allowed us to examine diurnal cycles, weekly patterns as well as spatially resolved long term changes. We combined a long-path Differential Optical Absorption Spectroscopy (DOAS) instrument with a cavity enhanced DOAS and applied several normalizations to the data sets in order to make the different measurement routes comparable. For the analysis of long term changes we used the entire unfiltered data set, for the comparison of spatial patterns we filtered out the accumulation of NO$_2$ when stopping at traffic lights for focusing on the changes of NO$_2$ spatial distribution instead of comparing traffic flow patterns, and for the generation of composite maps the diurnal cycle has been normalized by scaling the mobile data with coinciding citywide path-averaged measurement results.

An overall descending trend from 2010 to 2017 could be observed, consistent with the observations of the ozone monitoring instrument (OMI) and the Environment Protection Department (EPD) air quality monitoring network data. However, long term difference maps show pronounced spatial structures with some areas, e.g. around subway stations, revealing an increasing trend. We could also show, that the weekend effect, which for the most part of Hong Kong shows reduced NO$_2$ concentrations on Sundays and to a lesser degree on Saturdays, is reversed around shopping malls. Our study shows that the spatial differences have to be considered when discussing city-wide trends and can be used to put local point measurements into perspective. The resulting data set provides a better insight into on-road NO$_2$ characteristics in Hong Kong which helps to identify heavily polluted areas and represents a useful database for urban planning and the design of pollution control measures.



# 1 Introduction

Nitrogen dioxide ($NO_2$) is one of the major air pollutant and plays a key role in both tropospheric and stratospheric chemistry. It participates in the catalytic formation of tropospheric ozone ($O_3$) and also contributes to the formation of secondary aerosols (Jang and Kamens, 2001; Huang et al., 2014) and causes acid rain. High $NO_2$ concentration is known to be toxic to human. Nitrogen oxides ($NO_x$), defined as the sum of nitric oxide (NO) and $NO_2$, is released into the atmosphere from both natural and anthropogenic sources. Major sources of $NO_x$ include fossil fuel combustion, biomass burning, lightning and oxidation of ammonia (Bond et al., 2001; Zhang et al., 2003). In Hong Kong, vehicle emissions are the main source of $NO_x$. Similar to many metropolitan areas, a decreasing trend of ambient and roadside $NO_x$ levels has been observed (Carslaw, 2005; Keuken et al., 2009; Tian et al., 2011) which is contributed from the effective vehicular emission control measures in the past. However, the pollution levels measured at both ambient and roadside air quality monitoring stations are still occasionally exceeded the world health organization (WHO) guideline values of $40\,\mu g/m^3$ (annual) and $200\,\mu g/m^3$ (hourly) for $NO_2$, with more frequent exceedance of hourly $NO_2$ with high values observed at roadside stations. A rising trend of $NO_2/NO_x$ ratio with reduction of $NO_x$ is recorded at the roadside monitor stations in Hong Kong, which means the reduction rate of $NO_2$ is slower than NO in recent years (Tian et al., 2011). Vehicular $NO_2$ is either primarily emitted at the tail pipe or secondarily formed from oxidation of NO emission involving ozone and volatile organic compounds (VOCs) in the ambient (Muilwijk et al., 2016; Chang et al., 2016). The increase of $NO_2/NO_x$ ratio could either relate to the upgrades of vehicle engines and catalytic filters or changes in the composition and ambient level of VOCs. However, $NO_2$ concentration changes rapidly with time and has a very strong spatio-temporal variability, which is often unknown in urban areas (Longley et al., 2015). Regular roadside air quality monitoring stations are not sufficient to capture these variations and could not provide an overview of the roadside pollution situation representative for Hong Kong. Therefore, it is necessary to perform on-road mobile measurements for better understanding the pollutant distribution and spatial coverage of $NO_2$ for the entire city.

In order to capture the spatial and temporal variability of $NO_2$ concentrations in the central metropolitan area of Hong Kong, we use a combination of two different differential optical absorption spectroscopy (DOAS) techniques, a long path DOAS (LP-DOAS) and a cavity enhanced DOAS (CE-DOAS), as well as an ultraviolet (UV) based dual beam in-situ ozone monitor (Model 205, 2B Technologies). CE-DOAS is a relatively new spectroscopic measurement technique which uses an optical resonator to produce a long light path to enhance the absorption signal within a limited space (Platt et al., 2009). Sensitive measurements of trace gas have already been demonstrated by Langridge et al. (2006); Venables et al. (2006); Washenfelder et al. (2008); Thalman and Volkamer (2010); Min et al. (2016); Chan et al. (2017b). Compared to other in-situ $NO_2$ monitoring techniques, CE-DOAS is insensitive to other reactive nitrogen ($NO_y$) in the atmosphere, making it a better option for small spatial scale measurements and detection of spatial variation of trace gases. Its high accuracy allows fast sampling which is important for mobile measurements.

Mobile measurements are an effective tool to obtain the spatial and temporal variations of highly dynamic on-road pollutants. Therefore, it has been widely used for determining on-road vehicle emission factors (Vogt et al., 2003; Uhrner et al., 2007; Ning et al., 2012) and assessing the impacts of urban planning on air quality (Rakowska et al., 2014; Chan et al., 2017b).





Mobile CE-DOAS measurements of on-road $NO_2$ were performed in December 2010 and March 2017. The mobile measurements were used to investigate the relationship between on-road and ambient air quality. In addition, LP-DOAS measurements were performed to investigate the temporal variation of general ambient $NO_2$ in Hong Kong. Details of the mobile CE-DOAS and LP-DOAS experimental setups are presented in section 2. In section 3.2, the data filtering and normalization algorithms

applied to the mobile measurement data are introduced. The mobile $NO_2$ measurements are then analysed together with LP-DOAS and local monitoring station data for the long term trends, and results are shown in section 3.3. Section 3.4 presents the analysis of the characteristics of the weekend effect for different parts of the city. In addition, the spatial patterns of on-road $NO_2$ and the identification of pollution hotspots are presented in section 3.5.

## 2 Methodology

### 2.1 Mobile cavity enhanced DOAS

A CE-DOAS instrument was employed for mobile measurements using a sampling inlet positioned on top of the front part of the vehicle at a height of about 1.5 m above ground. The measurements were performed in December 2010 and March 2017 and divided into two parts, (a) measurement along a standard route that covers large part of the urban area in Hong Kong and (b) single measurement in different areas that are not covered by the route. The regular route covers Mong Kok, Central and

Causeway Bay which are the busiest areas in Hong Kong (see Figure 1). The standard route measurements were performed 2 to 3 times per day in order to cover non-rush hours, morning and evening rush hours. The varying route measurements were mostly performed during non-rush hours which aims to provide better spatial coverage and to identify pollution hotspots. Measurements performed in 2010 focus more on the on-road $NO_2$ spatial distribution and the identification of pollution hotspots. Therefore, the 2010 measurements include more non-standard route measurements to have a better spatial coverage. On the

other hand, the objectives of the 2017 measurements were refined to investigate the spatio-temporal variations over major pollution hotspots, that are mostly concentrated in the city center. As a result, we focused more on the standard route measurements over the city center in 2017.

The principle of the CE-DOAS (Platt et al., 2009) is similar to that of the cavity enhanced absorption spectroscopy (CEAS) (Fiedler et al., 2003). The measured absorption spectrum of an incoherent broad band light source (e.g. LED) is used to

determine the concentration of trace gases, which allows the application of the DOAS technique for the detection of multiple trace gases by a single instrument.

A schematic diagram of the CE-DOAS instrument is shown in Figure 2. The CE-DOAS consists of a blue LED light source, an optical resonator with two high reflective mirrors, a spectrometer and an air sampling system. Dielectric coated high reflective mirrors (reflectivity >99.98 % at 440 nm) are placed at both ends of the sampling cell to form an optical resonator.

Light from the high power blue LED (CREE XR-E royal blue, 440 nm - 455 nm FWHM) is coupled into the optical resonator by a convex lens with a focal length of 25 mm. Light escaped from the other side of the optical resonator is coupled to an optical fiber with a numerical aperture of 0.22 by a convex lens with a focal length of 50 mm and an aluminum mirror. The transmitted light is redirected to the spectrometer for spectral analysis through the optical fiber. Spectra are recorded by an



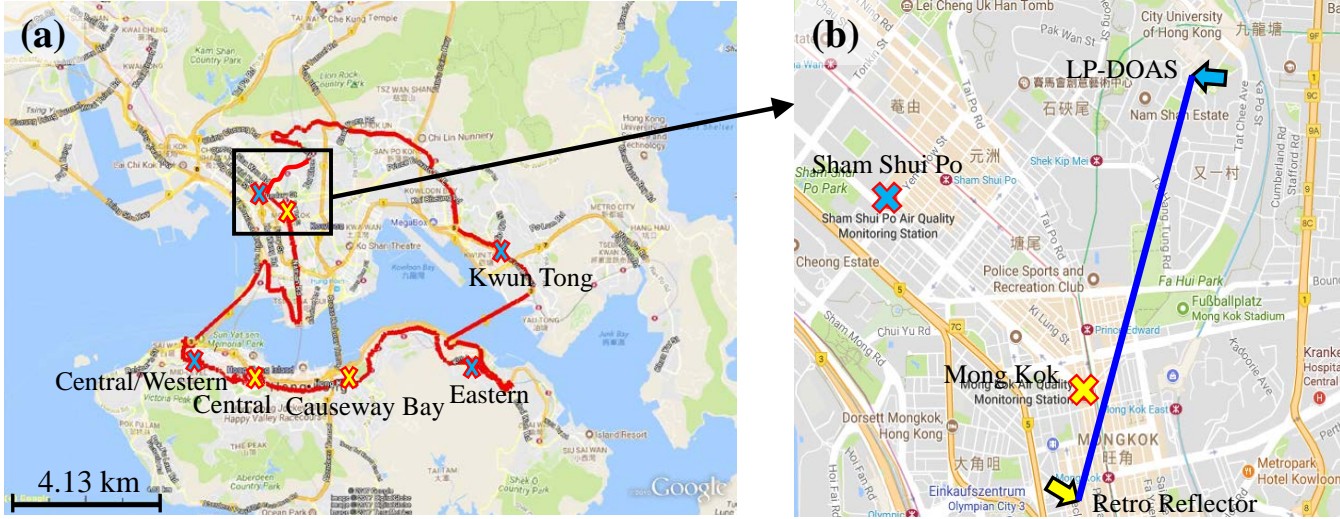

**Figure 1.** Map of Hong Kong city center. (a) The standard measurement route; (b) the location of LP-DOAS. Yellow crosses indicate 3 roadside EPD monitoring stations while blue crosses represent 4 ambient EPD monitoring stations. The blue line indicated in (b) represents the optical path of the LP-DOAS.

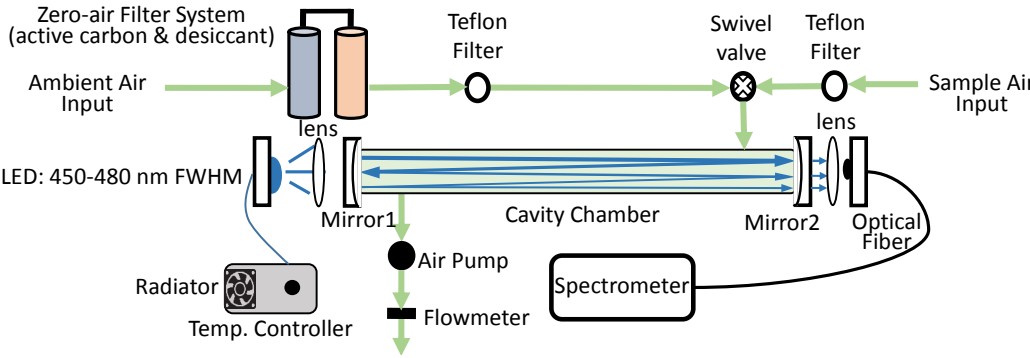

**Figure 2.** Schematic diagram of the experimental setup of the CE-DOAS.

Avantes spectrometer (AvaSpec-ULSi2048L-USB2) with a Sony ILX511 charge coupled device (CCD) detector. The spectral range of the spectrometer is adjusted to 395 nm - 492 nm with a spectral resolution of 0.47 nm (FWHM). The sampling cell is made by a Teflon pipe with length of 50 cm and a sampling volume of 286.3 cm$^3$. The sample flow of the system is achieved by a direct current vacuum pump located at the outlet side of the sampling chamber. A Teflon filter is placed in front of the inlet of the sampling cavity to avoid aerosols entering the sampling cavity and affecting the optical path by scattering and contamination of the high reflective mirrors. The time resolution of the CE-DOAS was adjusted to 4 seconds during the mobile measurement. Detailed description of the CE-DOAS instrument can be found in Platt et al. (2009); Chan et al. (2017b).



In this study, the software DOASIS (Kraus, 2005) was used for the CE-DOAS spectral evaluation. The CE-DOAS spectral fit is performed in the wavelengths from 435.6 nm to 455.1 nm, which includes several strong $NO_2$ and water vapor absorption bands. Reference absorption cross sections of $NO_2$ (Vandaele et al., 2002), $H_2O$ (Rothman et al., 2003), Glyoxal (CHOCHO) (Volkamer et al., 2005) and $O_4$ (Hermans et al., 1999) were included in the DOAS fitting.

## 2.2   Long path DOAS observations

A Light Emitting Diode (LED) based LP-DOAS system was installed on the roof top of the City University of Hong Kong building, providing measurement of near surface $NO_2$. The retro reflectors were placed on a high rise building located at the center of Kowloon, realizing an optical path of 1.9 km (total absorption path of 3.8 km). The spectral range of the spectrometer was adjusted from 400 nm to 462 nm with a spectral resolution of 0.4 nm (FWHM). The average altitude of the LP-DOAS light path is ∼50 m above ground level covering a long light path over the urban area of Hong Kong, providing representative measurements of ambient $NO_2$ level. Details of the experimental setup and the data retrieval procedure of the LP-DOAS can be found in Chan et al. (2012, 2017a). When focusing on the spatial variations, we used ambient $NO_2$ values measured by the LP-DOAS to normalize for the temporal dependency of the mobile CE-DOAS measurements. Since the mobile measurements record data from different parts of the city at different times of the day, the diurnal variability has to be normalized in order to produce a concentration map that is representable for daily average concentration of $NO_2$. Details of the normalization procedure are presented in section 3.2.

## 2.3   Local air quality monitoring network

Ambient $NO_2$ data in Hong Kong were acquired from the air quality monitoring network of Hong Kong which is operated by the Environmental Protection Department (EPD). The air quality monitoring network comprises 13 ambient and 3 roadside monitor stations (see Figure 1 for the locations of some of the stations). They cover both urban and rural areas in Hong Kong. The $NO_2$ and $NO_x$ concentrations are measured by in-situ chemiluminescence $NO_x$ analyzer. Ultra-violet (UV) absorption $O_3$ analyzer is used for $O_3$ monitoring. More details of the air quality monitoring network can be found on http://www.aqhi.gov.hk/en/monitoring-network/air-quality-monitoring-network.html. Hourly $NO_2$ concentrations from seven nearby air quality monitoring stations were used to compare to LP-DOAS and CE-DOAS $NO_2$ measurements. In addition, $NO_2$, $NO_x$ and $O_3$ data from the monitoring stations are used for long-term trend analysis. The reaction time of $NO$-$NO_2$-$O_3$ chemistry caused by traffic-induced turbulence in an inner city street canyon with high traffic density is about 12.6 seconds, as simulated by the regional atmospheric chemistry mechanisms (RACM) (VDI, 2017b) using typical values for photolysis frequencies and meteorological situaion. Since emissions accumulate for much longer than that, we can assume a equilibrium state with an average $NO_2/NO_x$ ratio of $0.70 \pm 0.22$ for our measurements was calculated, using our measured $O_3$ concentration with typical values for photolysis rate and an overcast sky (VDI, 2017a), derived from the Leighton ratio equation. This value could be confirmed by ratios derived from the EPD monitoring stations where NO measurements were available (see section 3.3).





## 2.4 OMI Satellite observations

The Ozone Monitoring Instrument (OMI) is a passive nadir-viewing satellite borne imaging spectrometer (Levelt et al., 2006) on board the Earth Observing System's (EOS) Aura satellite. The instrument consists of two charge-coupled devices (CCDs) covering a wavelength range from 264 nm to 504 nm. A scan provides measurements at 60 positions across the orbital track

covering a swath of approximately 2600 km. The spatial resolution of OMI varies from $\sim$320 km$^2$ (at nadir) to $\sim$6400 km$^2$ (at both edges of the swath). The instrument scans along 14.5 sun-synchronous polar orbits per day, providing daily global coverage observations.

In this study, NASA's OMI NO$_2$ standard product version 3 (SPv3) is used (Krotkov et al., 2017). The slant column densities (SCDs) of NO$_2$ are derived from Earth's reflected spectra in the visible range (402 - 465 nm) using an iterative sequential

algorithm (Marchenko et al., 2015). The OMI NO$_2$ SCDs are converted to vertical column densities (VCDs) by using the concept of air mass factor (AMF) (Solomon et al., 1987). The AMFs are calculated based on NO$_2$ and temperature profiles derived from the Global Modeling Initiative (GMI) chemistry transport model simulations with a horizontal resolution of 1° (latitude) $\times$ 1.25° (longitude) (Rotman et al., 2001). Separation of stratospheric and tropospheric columns is achieved by the local analysis of the stratospheric field over unpolluted areas (Bucsela et al., 2013).

## 3   Results and Discussion

### 3.1   NO$_2$ measurement comparison

Our LP-DOAS measurements of atmospheric NO$_2$ in Hong Kong started in December 2010. The data shows significant diurnal, weekly and seasonal variability. The daytime annual average NO$_2$ concentration measured by the LP-DOAS from 2011 to 2015 is 47.5 $\mu$g/m$^3$. A decreasing trend can be observed (see section 3.3 for more detailed discussion), but they all still higher

than the annual average of 40 $\mu$g/m$^3$ in WHO guideline (same standard as the Hong Kong air quality objective for NO$_2$). Additionally, episodes of high NO$_2$ levels are occasionally recorded, especially from long range transportation of air pollutants from mainland China (Kuhlmann et al., 2015).

A time series of NO$_2$ concentrations measured by LP-DOAS and EPD monitoring stations are shown in Figure 3. On one hand, both LP-DOAS and EPD measurements show similar variation pattern with higher values during daytime and

lower values at night. On the other hand, LP-DOAS and different EPD stations measurements also demonstrate different characteristics of NO$_2$. The significant spatial dependency of NO$_2$ is also confirmed in long-term changes (see section 3.3 for more detailed discussion). All measurements show an elevated NO$_2$ level during morning (8:00 to 10:00) and afternoon (17:00 to 19:00) rush hours. However, the absolute concentration measured by different stations varies in a wide range. In addition, differences in measurement height also contribute to differences among these measurements. In order to have a better overview

of the NO$_2$ spatial distribution, temporal variation and their emission source pattern, we performed mobile measurements of on-road NO$_2$ using a CE-DOAS instrument. On road measurement can easily be influenced by the traffic condition, e.g., accumulation of emission during traffic congestion, and the diurnal variation of ambient NO$_2$. In order to correct for these





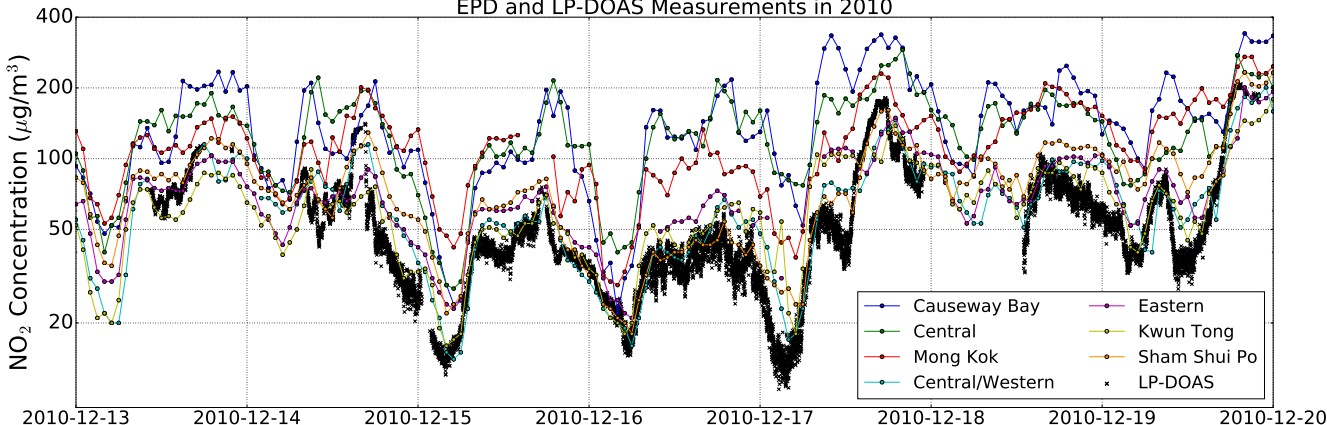

**Figure 3.** Time series of NO$_2$ concentration measured by LP-DOAS and EPD Monitoring stations during the measurement campaign in 2010.

effects in the mobile measurement, we have filtered data which is influenced by traffic condition and normalized the on-road measurement for diurnal variation of NO$_2$.

## 3.2 Data filtering and normalization

### 3.2.1 Comparison of concentrations during fluent traffic and traffic congestion

5   Traffic congestion can result in higher pollution levels due to accumulation of vehicle emissions, caused by less turbulent mixing with cleaner air and longer NO to NO$_2$ reaction time. It has been observed that high concentrations of NO$_2$ were recorded during low speed driving in our measurements, i.e. in a traffic jam or waiting in front of a traffic light. Figure 4 shows the time series of vehicle speed and measured NO$_2$ concentration during a traffic congestion on $2^{nd}$ Mar 2017. Note that the vehicle speed is calculated from the GPS data with an error about 0.6 m. Converting the error into vehicle speed would be

10   1.4 km/h. Therefore, the vehicle speed is never zero even if the vehicle stops. In the example shown in Figure 4, the vehicle slowed down and stopped for half a minute at a traffic light. The NO$_2$ level goes up from about 100 $\mu$g/m$^3$ to more than 400 $\mu$g/m$^3$. The NO$_2$ level rises about 8 s after the vehicle stopped. When the vehicle started moving again, the measured NO$_2$ level gradually dropped back to the pre-stop level within 20 s.

  In order to separate data that is influenced the NO$_2$ spikes induced by traffic congestion or idling, we filtered data from 8 s

15   after the vehicle speed drop below 5 km/h to 20 s after the vehicle speed goes above 5 km/h again. In order to avoid filtering data due to poor GPS signal, this filter only applies when the vehicle speed is below 5 km/h for more than 8 s. The average NO$_2$ concentrations for standing condition are 239 $\mu$g/m$^3$ which is 14.5 % higher on average. We filter out traffic light or traffic jam stops only to have a consistent NO$_2$ spatial distribution under fluent driving condition for the direct comparison of two measurements in different days and years, in order to focus on the concentrations instead of the stopping frequency. This filter





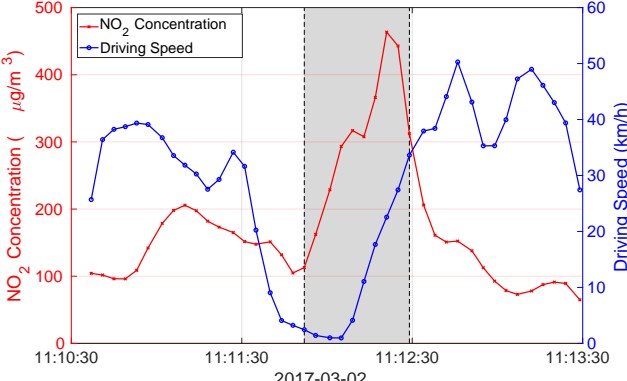

**Figure 4.** Time series of the driving speed and the coinciding $NO_2$ concentration during stops due to traffic congestion. Data in gray area will be filtered out in later analysis.

criterion removed 37 % and 30 % of the total number of measurement data in 2010 and 2017. However, since the filter mainly removes measurements at low speed or standing, only 10 % and 11 % of the spatial points were removed for 2010 and 2017, respectively.

### 3.2.2  Normalization of the diurnal cycle

In order to separate the $NO_2$ spatial and temporal variability and show a representative spatial distribution of $NO_2$ in Hong Kong, we developed an algorithm using LP-DOAS measurements to normalize for the diurnal variations. Although the LP-DOAS measurement covers a long light path over the urban area in Hong Kong, the $NO_2$ values provided might still not be representative for all measurement areas due to local influences. Therefore, we use a normalized long term average of diurnal $NO_2$ cycle for each weekday to correct for the temporal variation effect. It is less depending on outliers caused by the overpass

pollution plume and can also interpolate data gaps due to instrumental problems and bad weather.

LP-DOAS measurements of atmospheric $NO_2$ for each day are first normalized by dividing by the daily mean $NO_2$ concentration. The resulting normalized $NO_2$ level are then averaged for each day of the week over a period of 2 years to obtain a representative diurnal $NO_2$ variation pattern. The normalized and averaged diurnal $NO_2$ variation pattern of the corresponding weekday is scaled and shifted to fit the normalized LP-DOAS measurement for each day during the mobile measurement

campaign. The inverse of the 1 $\sigma$ (standard deviation) variation of the 2-year averaged and normalized $NO_2$ level is used as weighting in the least squares regression to scale and shift the long term average diurnal pattern. In order to avoid single high value affecting the whole regression, normalized $NO_2$ level exceeded the 1 $\sigma$ variation of the 2-year averaged and normalized $NO_2$ level were not considered in the regression process. Figure 5 shows the normalized $NO_2$ concentration measured by the LP-DOAS on $17^{th}$ Dec 2010. Normalized 2-year Friday mean $NO_2$ diurnal pattern, the diurnal pattern of scaled $NO_2$ measure-

ment taken on $17^{th}$ Dec 2010 and normalized EPD monitoring data are shown as well. All data illustrate similar characteristics





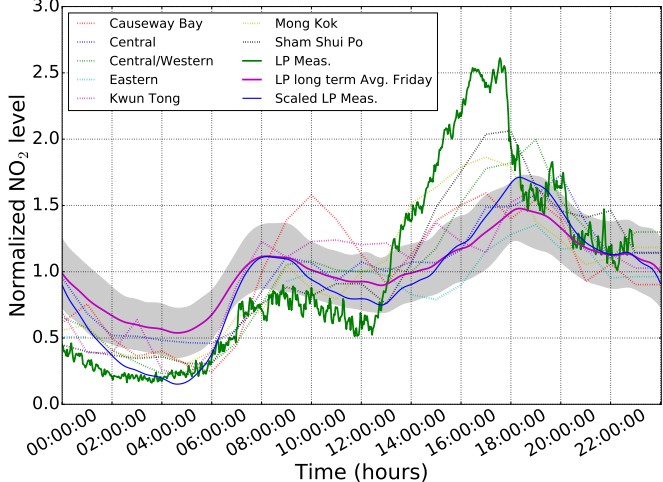

**Figure 5.** Normalized diurnal cycle of $NO_2$ on Friday in Hong Kong in 2010 measured by the LP-DOAS and EPD monitoring stations. EPD measurements on $17^{th}$ Dec 2010 from 7 monitoring stations are indicated as dashed line. The green curve represents LP-DOAS measurement while purple line is the 2 years averaged diurnal pattern with shadowed area of the $1\,\sigma$ standard deviation variation. The blue line shows the scaled and shifted diurnal pattern of ambient $NO_2$ on Friday, $17^{th}$ Dec 2010.

with significant peaks in the morning (8:00 to 10:00) and evening (17:00 to 19:00) rush hours. The fitted long term diurnal pattern is then used to correct for the diurnal effect of the mobile measurement. Mobile measurements are multiplied by the simultaneous $NO_2$ level of the resulting normalized LP-DOAS diurnal pattern to obtain a more representative value for the measurement areas.

## 3.3 Long-term trends of $NO_2$

On road CE-DOAS measurements are analyzed together with LP-DOAS and EPD monitors data to investigate the long term trend of on-road and ambient $NO_2$. The observed trends at different locations are compared to the changes of the mobile on-road CE-DOAS $NO_2$ measurements in 2010 and 2017 taken within 100 m radius of the 3 EPD roadside stations or within 1 km radius of the center of the LP-DOAS measurement path (Figure 6a, b, c and d). The time series represent monthly averaged ambient $NO_2$ concentrations measured during daytime. OMI satellite observations of monthly average tropospheric $NO_2$ VCDs over Hong Kong are shown in Figure 6e. The data were filtered for cloud fraction larger than 50 % and averaged for OMI pixel within 50 km of the measurement site. On-road, ambient, and satellite measurements of $NO_2$ all show a decreasing trend. Ambient $NO_2$ levels measured by the LP-DOAS show a descending trend with a rate of 2.5 % per year. Stronger decreasing trends of roadside $NO_2$ are observed by EPD in Mong Kok, Causeway Bay and Central roadside station with annual decreasing rates of 4.4 %, 3.3 % and 4.8 %, respectively. A similar reduction rate is also observed by on-road CE-DOAS measurements. Comparing the CE-DOAS measurement taken in 2010 and 2017, on-road $NO_2$ levels are overall reduced by 28 % for areas



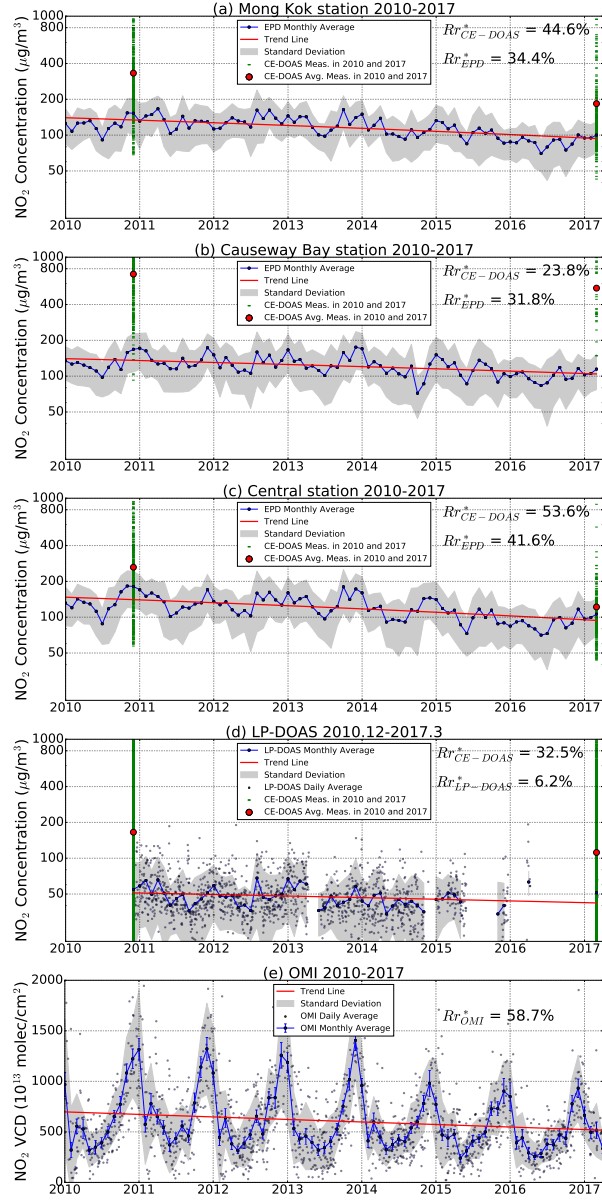

**Figure 6.** Monthly averaged daytime NO$_2$ concentration from Jan 2010 to Mar 2017 measured by three EPD stations and LP-DOAS. Red dots indicate the averaged NO$_2$ concentration measured by the CE-DOAS within 100 m radius of (a) Mong Kok, (b) Causeway Bay and (c) Central roadside station. (d) shows the CE-DOAS measurements within 1 km radius of the center of the LP-DOAS measurement path and monthly daytime averaged ambient NO$_2$ levels observed by the LP-DOAS. (e) shows the monthly averaged OMI tropospheric NO$_2$ VCDs over Hong Kong. The reduction rates Rr indicated on the figures are calculated by taking the relative difference between averaged data taken in December 2010 and March 2017. Low reduction of LP-DOAS is due to sparse measurements in March 2017.





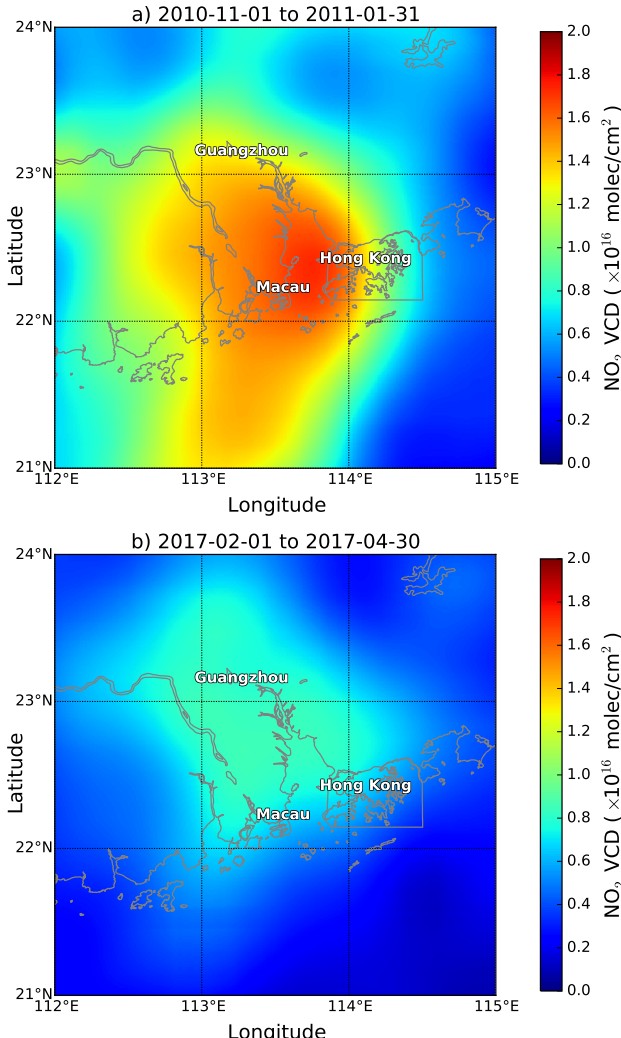

**Figure 7.** Averaged OMI tropospheric $NO_2$ VCDs over Pearl River Delta a) from $1^{st}$ of Nov 2010 to $31^{st}$ Jan 2011 and b) from $1^{st}$ Feb 2017 to $31^{st}$ Apr 2017.





along the standard measurement route which would correspond to an annual decreasing rate of 4.0 %. $NO_2$ levels in 85 % of the measurement area are significant reduced (>1 ppb) by 37 % on average, whereas $NO_2$ levels in 14 % of the area are elevated (>1 ppb) by 22 % on average. The reduction rate for on-road $NO_2$ levels around the EPD roadside monitor stations varies from 24 % to 54 %. This reduction change can also be observed from space by OMI satellite. Tropospheric $NO_2$ VCDs show a

descending trend with a rate of 3.7 % per year. In addition, Figure 7a and b show tropospheric $NO_2$ VCDs over the Pearl River Delta from $1^{st}$ of Nov 2010 to $31^{st}$ Jan 2011 and from $1^{st}$ Feb 2017 to $31^{st}$ Apr 2017, respectively. In general, tropospheric $NO_2$ VCDs are reduced by ∼50 % (7 % per year) over Hong Kong, while the reduction over Pearl River Delta is ranging from 30 - 60 %.

Averaged on-road $NO_2$ concentrations measured along the standard route during December 2010 and March 2017 are shown

in Figure 8a and b, the differences in Figure 8c. The measurement routes are slightly different due to road constructions and maintenance. In general, a significant reduction (ranging from 20 % to 50 %, and on average 4 % per year) of on-road $NO_2$ can be observed which is consistent with the LP-DOAS and EPD monitor data. The reduction of on-road $NO_2$ level along Nathan Road, the busiest road in Kowloon, is ranging from 50 % to 60 % (around 7 % to 8 % per year). On the other hand, an enhancement of $NO_2$ level can be observed around subway stations, e.g., Hong Kong University station, Kwun Tong

station, Diamond Hill station, Ngau Tau Kok station, etc. It probably reflects the fact that there are more bus terminals or bus stops surrounding metro stations in 2017 compared to 2010. Data from the transport department shows that the total number of licensed franchised bus has slightly increased by 3 % from 5729 in 2010 to 5916 in 2016 (http://www.td.gov. hk/en/transport_in_hong_kong/transport_figures/monthly_traffic_and_transport_digest/index.html). Although the number of franchised bus only has a small contribution to the total number of vehicle in Hong Kong (608 thousands in 2010, 746 thousands

in 2016), franchised buses can account for up to 40 % of the traffic at busy traffic corridors (http://www.info.gov.hk/gia/general/ 201512/31/P201512310204.htm). Average daily public transport usage also increased from 11.6 millions time per day in 2010 to 12.6 millions time per day in 2016. According to the annual reports of Transport International Holdings Limited, the parent company of the Kowloon Motor Bus Company, the largest franchised bus operator in Hong Kong, the total number of buses running by the company increases slightly from 3988 in 2010 to 4162 in 2016. However, the number of bus routes reduced from

393 in 2010 to 384 in 2016. These changes are mainly due to the reformation of the operational strategies of the franchised bus operators. Due to the expansion of the metro system in Hong Kong, the role of bus has gradually changed from point to point long distance services to a connector between destination and metro stations. Therefore, enhancement of $NO_2$ levels is observed around metro stations. On the other hand, franchised bus operators in Hong Kong started to introduce low emission buses (i.e., Euro IV and V) since 2009. Buses with model earlier than Euro III are proven to be more polluted than latest models

(Dallmann et al., 2011; Mock, 2014; Lau et al., 2015; Pastorello and Melios, 2016). Therefore, bus companies started to install retro fit catalytic convertor on earlier bus models and these buses will be replaced completely by buses with higher emission standard by 2021. In addition, the government has set up franchised bus Low Emission Zones (LEZs) in three busiest traffic corridors in Hong Kong on $31^{st}$ Dec 2015. Buses with emission standard below Euro IV are not allowed to operate within these low emission zones. Therefore, both roadside and its nearby ambient $NO_2$ levels show a descending trend. Navigation

(water transport), road transport and public electricity generation are the largest sources of $NO_x$ according to the 2015 Hong



**Figure 8.** Averaged on-road NO$_2$ concentrations measured along the standard route during (a) December 2010 and (b) March 2017. (c) shows the relative differences between 2010 and 2017. The markers indicate the location of metro stations.





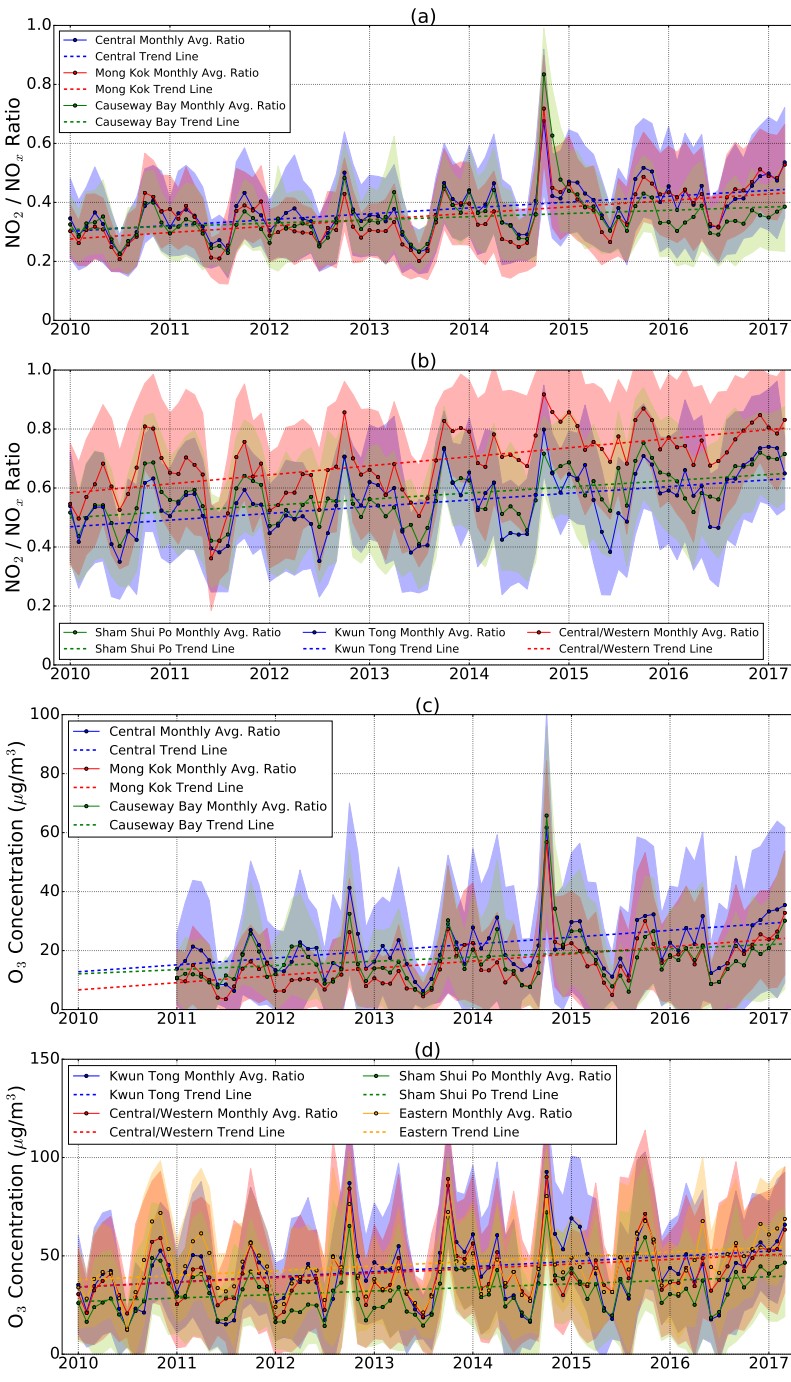

**Figure 9.** Monthly averaged NO₂/NOₓ ratio from EPD (a) roadside stations (b) ambient stations. O₃ concentrations measured by EPD (c) roadside stations (d) ambient stations are shown. Shadowed area indicates the 1 $\sigma$ standard deviation variation of measurements.





Kong Emission Inventory Report and data from the EPD Hong Kong Air Pollutant Emission Inventory, accounting for 33 %, 30 % and 25 % of total $NO_x$ emissions in 2010 (only $NO_x$ is available in the EPD inventory) and 37 %, 18 % and 28 % in 2016, respectively. $NO_x$ emissions from navigation and public electricity generation are rather constant, while emissions from road transport show a significant reduction of ∼50 % from 32.1 tonnes in 2010 down to 16.2 tonnes in 2015. This is coherent with the decreasing trend of $NO_2$ from 2010 to 2017. We have looked into the $NO_2/NO_x$ ratio as well as the $O_3$ concentration in order to better understand the impacts of reduction of vehicular emission of $NO_x$. An increasing trend of $NO_2/NO_x$ ratio is observed from both roadside and ambient monitoring stations. Figure 9 shows the $NO_2/NO_x$ ratio for (a) roadside and (b) ambient stations. Ozone concentrations from both (c) roadside and (d) ambient stations are shown for reference. Decreasing roadside $NO_2$ level with increasing $NO_2/NO_x$ ratio implies a significant reduction of primary NO emissions. The reduction of primary NO is could be subjected to the upgraded catalytic converter of diesel vehicles (from Euro III or earlier model to Euro IV and V) which reduces the total $NO_x$ emission and increases the $NO_2/NO_x$ ratio (Kašpar et al., 2003). Newer diesel engines in general reduce the total $NO_x$ emission by ∼50 % according to the European emission standards for diesel passenger cars (EU emission standards, 2007). The Euro III diesel engines emission limit of $NO_x$ is 0.50 g/km, whereas the Euro IV emissions limit has reduced half to 0.25 g/km. However, this standard might not fully reflect the real driving condition (Franco et al., 2014) and it should be confirmed by more realistic mobile measurements. Furthermore, Tian et al. (2011) observed a rising roadside $NO_2/NO_x$ ratio as well coincided with the introduction of new environmental friendly pre-Euro light and heavy duty vehicles in 2000 and 2003. Ning et al. (2012) also suggested that the proposal of replacing Euro II and III franchised buses to meet Euro IV or even higher emission standards will result in an increase of roadside $NO_2/NO_x$ ratio. In addition, a general rising trend of ambient and roadside ozone is also observed from the EPD monitoring data. The increase of atmospheric $NO_2/NO_x$ with large reduction of NO may have been contributed to the recent increase of $O_3$ level in the roadside stations, as less NO is available for the titration process under heavy $NO_x$ environment.

## 3.4 Weekend effect

Figure 10a shows the five years average diurnal cycle of $NO_2$ of each day of the week measured by LP-DOAS, and the seven years average $NO_2$ diurnal pattern measured by EPD Sham Shui Po, Mong Kok and Causeway Bay station are shown in Figure 10b, c and d, respectively. The diurnal pattern of $NO_2$ illustrates different characteristics between weekdays and weekend. Different measurement locations also show different characteristics of $NO_2$ during weekend. The LP-DOAS measurement indicates the $NO_2$ concentration is on average 3.3 % lower on Saturday and 8.7 % lower on Sunday compared to weekdays. However, the morning rush hour (8:00 to 10:00) peak of $NO_2$ is significantly reduced by 23.1 % on Sunday, while the evening rush hour (18:00 to 20:00) peak shows a less pronounced reduction of 9.7 %. $NO_2$ measurements from the Sham Shui Po ambient station and the LP-DOAS show similar diurnal variation pattern and weekend reduction. The weekend reduction is less pronounced for the roadside measurements in Mong Kok, the $NO_2$ level is on average 3.8 % lower on Sunday compared to weekdays, with reduction during the morning and evening rush hours of 13.1 % and 7.3 %, respectively. Similar weekend reductions are also observed by other EPD roadside stations, i.e., Causeway Bay and Central. These differences in the diurnal cycle are most likely due to different types of land use. Traffic emissions are the main source of $NO_2$ in urban areas which





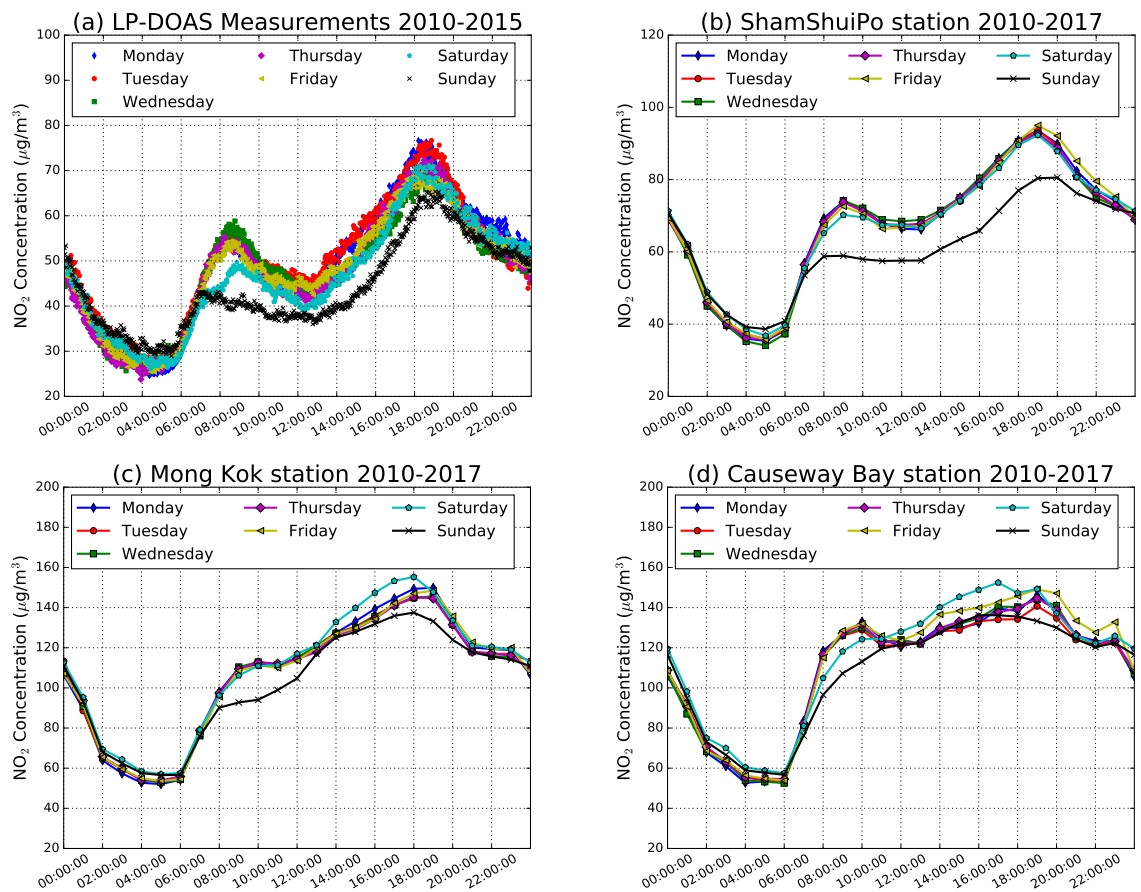

**Figure 10.** (a) shows the five years average diurnal cycle of NO$_2$ of each day of the week measured by LP-DOAS.The seven years average diurnal cycle of NO$_2$ of each day of the week measured by EPD (b) Sham Shui Po station, (c) Mong Kok station and (d) Causeway Bay station.



**Figure 11.** Mobile CE-DOAS measurement of on-road NO₂ on (a) Monday (6$^{th}$ Mar 2017) and (b) Sunday (5$^{th}$ Mar 2017). Coinciding NO₂ concentration measured by the 7 EPD stations are shown on the map as circle markers using the same color scale for the concentrations and differences. (c) shows the differences between Monday and Sunday. The markers indicate the location of major shopping malls.

is strongly dependent on human activities. In residential areas, traffic is reduced during weekend as most of the residents do not work on Sunday, e.g., frequency of buses is reduced during weekend. However, the traffic load is mostly unchanged in commercial areas, since shops are open as well on Sunday.



In order to further investigate the relationship between residents' activities during weekdays and weekend and $NO_2$ emissions, we have looked into the morning standard route measurements on a sequential Sunday and Monday in 2017. Two sequential days are used for comparison so as to avoid influences from different meteorological conditions. $NO_2$ concentration maps measured on Sunday and Monday are shown in Figure 11a and b and its difference in c. The $NO_2$ level on Sunday is on

average about 45 % lower than that of Monday. The mobile measurements are in general agreement with coinciding EPD data, while discrepancies can be observed for peak values captured by the more frequently measuring CE-DOAS. This discrepancy is mainly due to the difference in measurement time. Mobile measurement recorded the instant concentration of on-road $NO_2$ which could easily be influenced by a single incident, especially the on-road $NO_2$ level varies rapidly. On the other hand, EPD monitors provide hourly averaged $NO_2$ concentration which tend to average out those local pollution peaks. Besides, four

EPD ambient monitoring stations are located more than 15 m above ground level. Therefore, EPD ambient stations are expected to measure lower $NO_2$ concentrations compared to on-road CE-DOAS measurements. In addition, $NO_2$ concentrations of each location show rapid changes which are highly dependent on the traffic flow. However, a consistent elevated $NO_2$ level is observed over the most busy roads, such as Nathan Road in Kowloon, western and eastern Harbor Cross Harbor tunnels. 85 % of the measurements show significant higher (>1 ppb) $NO_2$ concentrations, whereas 13 % of the measurements show

significant lower (>1 ppb) $NO_2$ concentrations on Monday compared to Sunday. The spatial pattern of elevated $NO_2$ level on Sunday matches with the location of large shopping malls. Similar difference maps between other workdays and Sunday are observed. The number of licensed private car grows by ∼30 % from 415 thousands in 2010 to 536 thousands in 2016, while the public transport usage increases by ∼9 % from 11.6 millions time per day in 2010 to 12.6 millions time per days in 2016 (http://www.td.gov.hk/en/transport_in_hong_kong/transport_figures/monthly_traffic_and_transport_digest/index.html). These

numbers imply that there is a significant increase of weekend drivers in Hong Kong. People are taking public transport for daily commute to avoid traffic jam in the weekday, while go out to shopping with their own car in the weekend. As the parking spaces are limited around these shopping areas and results in low speed cruising and traffic congestion around these major shopping areas during weekend. As a consequence, an enhancement of $NO_2$ level can be observed over these locations. This is an interesting example of how people's daily life influences the pollution patterns.

## 3.5   Spatial distribution of $NO_2$ in Hong Kong

In order to have a better overview of major pollution hotspots in Hong Kong, all measurements taken in 2010 were spatially averaged to a high resolution grid of $20\,m \times 20\,m$ (Figure 12 (a)). These measurements covered most of the major roads in Hong Kong, including highway, urban, sub-urban and rural area. As the spatial coverage of measurements taken in 2010 and 2017 is quite different and there is a general decreasing trend of $NO_2$, we only use data measured in 2010 for the spatial

distribution analysis to avoid any bias toward lower value over the city center. Elevated $NO_2$ levels are mainly distributed over motorways and busy roads that always with high traffic intensity in the city center, e.g. No. 8 and No. 9 motorway, Nathan Road in Kowloon, Queen's Road in Central, and Hennessy Road from Admiralty to Causeway Bay. About 29 % of the on-road measurements exceeded the WHO one hour guideline value of $200\,\mu g/m^3$, while 27 % of the data measured in the city center exceed the guideline. High $NO_2$ values over motorways are probably due to having more heavy-duty vehicles. On the other



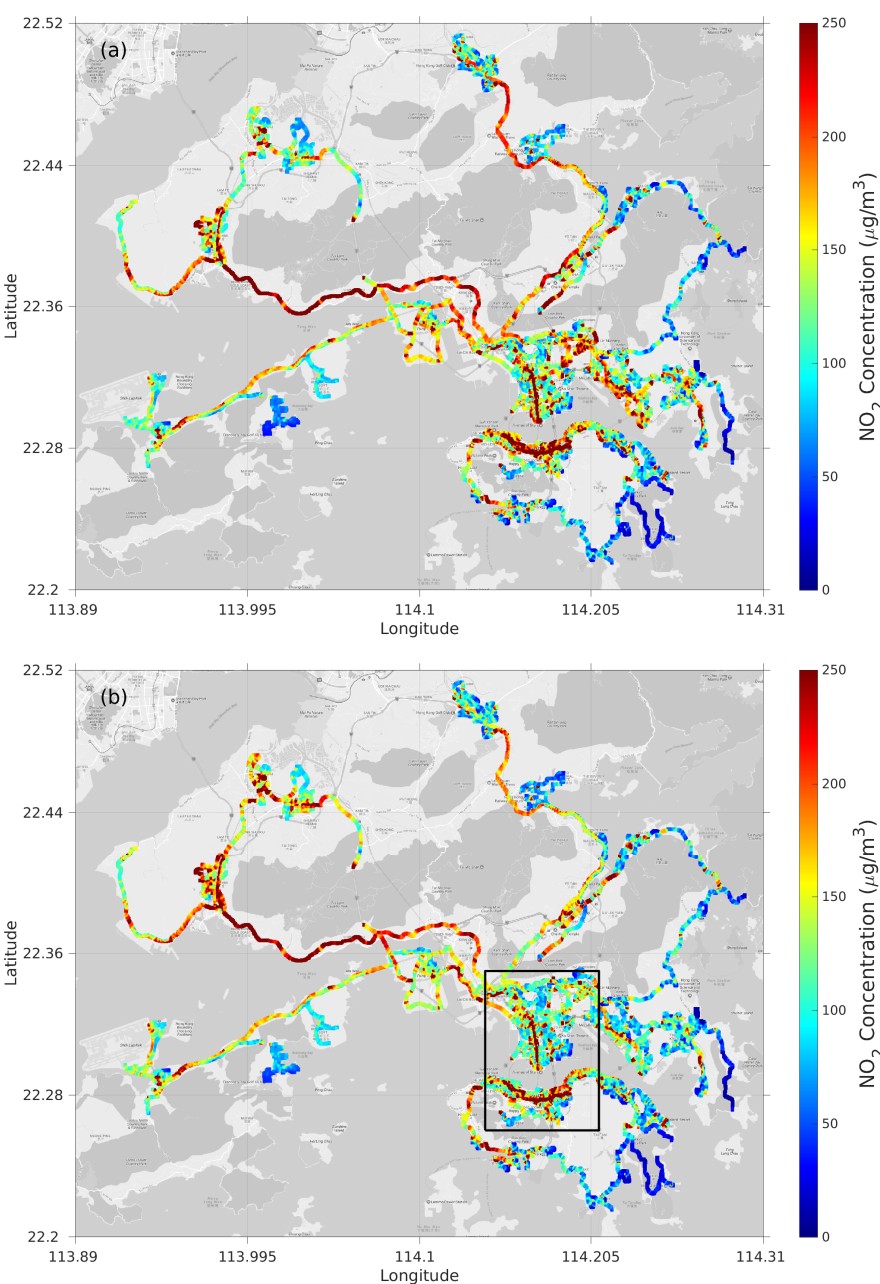

**Figure 12.** (a) Spatial distribution of NO$_2$ in Hong Kong measured by the mobile CE-DOAS in 2010. (b) Normalized spatial distribution of NO$_2$ over Hong Kong measured by the mobile CE-DOAS in 2010. The CE-DOAS data is normalized using coinciding normalized LP-DOAS data. The black box indicates the area of the city center used for the other maps.





hand, traffic congestion and street canyon effects (Rakowska et al., 2014) are the major cause of elevated on-road $NO_2$ in the city center.

As described in section 3.2.2, on-road pollutants mainly produced by vehicles and the traffic flow patterns also have a large impact on pollutant distributions (Westerdahl et al., 2005; Kaur et al., 2007; Huan and Kebin, 2012; Rakowska et al., 2014; Fu
et al., 2017). The diurnal dependency of the measurement times is corrected for using the simultaneous normalized LP-DOAS measurement. The normalized spatial distribution of on-road $NO_2$ is shown in Figure 12(b). This normalized dataset is now representative for the daily average. $NO_2$ levels over some regions are significantly enhanced after applying the normalization, particularly, the residential area in Yuen Long district and Tung Chung district, where the Hong Kong International airport is located. Some other areas (mainly at the city center and highways) obtained lower $NO_2$ values after normalization. Enhance-
ment of $NO_2$ concentrations after normalization for certain areas is due to the fact that the mobile measurement took place during non-peak hours during the day, while reduction of $NO_2$ concentrations is due to the measurement vehicle overpassing the regions during rush hours of the day. Compared to unnormalized data, only 27 % of normalized on-road measurements exceeded the WHO one hour guideline and about 20 % of the area in the city center exceed the guideline. The slightly decreased $NO_2$ level in both all over Hong Kong and city center are presumably due to the fact that the measurement campaigns are
conducted during daytime when the $NO_2$ level is in general higher compared to nighttime.

## 4   Summary and conclusions

A high resolution spatial distribution map of street level $NO_2$ makes identifing city pollution hotspots possible. It could meanwhile provide valuable information for urban planning as well as help with the development of pollution control measures. For obtaining the pollutant information, on-road mobile CE-DOAS measurements were successfully deployed in Hong Kong
in December 2010 and March 2017, respectively. The diurnal dependency due to the different sampling time of mobile measurements was normalized through combining the continuous measurements of LP-DOAS. Furthermore, the algorithm, which was developed to separate and filter the accumulation of local emissions due to traffic congestion, helped us focusing on the concentrations instead of the stopping frequency while the maps' comparison.

The long term trend and spatial variations of ambient, roadside and on-road $NO_2$ levels were investigated by analyzing on-
road CE-DOAS measurements together with LP-DOAS and EPD monitor stations. The long term trend analysis showed that the ambient $NO_2$ level was descending with a rate of 2.5 % per year, while the roadside $NO_2$ level showed a strong decreasing trend with annual reduction rate ranging from 3.4 - 4.9 %. This observation matched with the mobile measurement results that on-road $NO_2$ was in general reduced by 20 - 50 % between 2010 and 2017. The changes of the operational strategies of the major franchised bus company in Hong Kong could be revealed by the enhancements of $NO_2$ level observed at locations close
to metro stations. In addition, a rising trend of $NO_2/NO_x$ ratio was observed in both roadside and ambient monitor data. This was mainly subjected to the reduction of vehicle emissions which typically associated with $NO_2/NO_x$ ratio.

The temporal emission characteristic of different districts in Hong Kong were investigated using mobile measurements taken on different days of the week. The weekend reduction rate of on-road measurements was much higher than the long term



ambient / roadside observation of LP-DOAS and EPD monitoring stations. By analyzing the spatial pattern of the weekend reduction effect, we found that the $NO_2$ levels of most residential districts were reduced on Sunday while commercial and shopping areas showed a rather constant $NO_2$ level throughout the week. The mobile CE-DOAS measurements presented in this paper offered a full-scaled perception for the on-road $NO_2$ characteristics in Hong Kong. Simultaneously, these spatial distribution measurement results are also important for chemical transport model validations and assessment of human health effects.

*Competing interests.* The authors declare that they have no conflict of interest.

*Acknowledgements.* The work described in this paper was jountly supported by the German Academic Exchange Service (DAAD) Programme des Projektbezogenen Personenaustauschs (PPP) (project ID: 57334317), the Germany / Hong Kong Joint Research Scheme sponsored by the Research Grants Council of Hong Kong and the German Academic Exchange Service (Reference No. G-CityU104/16) and the Research Grants Council of the Hong Kong Special Administrative Region, China (Project No. CityU 11305817). We thank Annette Schütt, Teng Fei, Song Hao, Willy Ying for helping with the organization of the measurement campaign.





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
