# Peer review of "Analysis of spatial and temporal patterns of on-road NO2 concentrations in Hong Kong"

_Atmospheric Measurement Techniques, 2018_

## Referee Comment (RC1) · Anonymous Referee #1 · 12 Sep 2018

The manuscript 'Analysis of spatial and temporal patterns of on-road NO2 concentrations in Hong Kong' presents an investigation of the spatial and temporal variability of street level NO2 concentrations in Hong Kong. Two on road measurement campaigns were performed in 2010 and 2017 which combined both remote sensing LP-DOAS and mobile in-situ CE-DOAS measurements. As the measurements were taken at different time under different conditions, the authors put a big effort on filtering and normalizing the data in order to make these data sets comparable. Details of the filtering and normalizing procedure are presented. The filtered data were used to analyze the long term trend as well as the temporal and spatial characteristic of NO2. The authors also show some interesting characteristics, e.g., enhancement of NO2 levels around shopping area during weekends and increase of NO2 around metro stations. The manuscript

is in general well written and the topic is of interest of the general audiences of "Atmospheric Measurement Techniques". However, some clarifications are necessary. I suggest this manuscript for publication after a minor revision.

Specific comment:

As the measurement campaign were only performed in 2010 and 2017 each for a week. Although the authors have put a lot of effort in filtering and normalizing the data, it is difficult to say the measurements are representative for the general condition. I understood that it is difficult to have longer term measurements, but the authors should at least clarify that it is only a short term measurement and avoid using the term 'representative'. In addition, the measurement campaigns were performed in different seasons of the year and the meteorological conditions could be quite different. Is there any good reason?

Section 3.2.1: The authors present an example of the NO2 level under congestion condition and concluded that to filter data take 8 s after the vehicle speed below 5km/h. The lag time shown in this case is the combination of accumulation of NO2 in ambient plus the lag time of the instrument. The lag time of the instrument is fixed and can be measure, while the time of accumulation of NO2 varies. Clarification is necessary.

Figure 4 caption: Change 'during stops due to traffic congestion' to 'during traffic congestion'.

Section 3.3: Deriving long term trend from 2 weeks of measurements is not very convincing. The authors should state clear the purpose of comparing these short term measurements. The analysis of EPD and LP-DOAS data is variable though.

Figure 6: This plot contains a lot of information already. However, I still would like to know whether it is possible to compare the EPD data measured at the same time when the CE-DOAS was passing by and how's the correlation in between? The labels in the plots should be larger. The date in the title of each plot is redundant, please remove

[Figure]

them.

Figure 8: How does the average map calculated for each year? Does it corrected for the diurnal effect? Since the authors described the measurements were taken during the different time of the day which contains the morning and evening rush hours and non-rush-hour at noon, it may lead to a bias in averaging all measurements.

Figure 11: $NO_2$ concentration measured by the 7 EPD stations are shown on the map as well. But the authors don't describe any results such as the difference between EPD measurements and the coinciding closest on-road measurements, and the $NO_2$ average changes in these 2 years of EPD stations.

Page 5 line 28: 'a equilibrium state' to 'an equilibrium state'.

Page 12 line 2- 3: Explain why >1 ppb is significant. I suppose this is related to the detection limit of the instrument. Please specify it in the methodology section.

Page 15 line 10: 'primary NO is could be. . .' is grammatically incorrect. Please revise.

Page 17 line 3: If the traffic load is mostly constant in commercial areas which include most shopping malls on Sunday, why the differential map shows the decrease of $NO_2$ around shopping malls? A better description is necessary.

Page 18 line 14 and 15: Same as before, explain why >1 ppb is significant.

Page 20 line 20: I couldn't see the causal relationship between the increase amount of private cars and public transport usage with the significant increase of weekend drivers in Hong Kong. The authors should describe it better.

---

## Referee Comment (RC2) · Anonymous Referee #2 · 30 Oct 2018

In this manuscript, Zhu et al. report on measurements of boundary layer NO2 in Hong Kong using different techniques. In two campaigns, car-based measurements with a CE-DOAS instrument were performed for several days at different times of the day, covering both rush-hour and normal conditions. These measurements are complemented by data from the in-situ measurement network, a long path DOAS instrument operating during and in between campaigns, and OMI satellite data. Data were analysed for their temporal trend, the diurnal profile, the week-end effect, their spatial distribution and the NO2 / NOx ratio.

The paper reports interesting measurement results from a highly polluted city enforcing strict emission controls and highlights some nice local effects such as changes in pollution levels around metro stations. The manuscript is overall well written but fo-

cuses on reporting measurement results and a qualitative interpretation. It therefore does not fit well into the scope of AMT ("The main subject areas comprise the development, intercomparison, and validation of measurement instruments and techniques of data processing and information retrieval for gases, aerosols, and clouds.") but should rather have been submitted to ACP in my opinion. It would also benefit from a more quantitative discussion including error bars.

Nevertheless, I recommend it for publication after the following points have been fully addressed.

1. Was any correction applied to the in-situ chemiluminescence NOx analysers for cross-sensitivities?

2. I'm not yet convinced by the discussion of the NO2 to NOx ratios. While I can understand that the ratio is driven by the fraction of NOx emitted as NO close to the source, and therefore a change in technology used in the car fleet can have an impact on NO2 to NOx ratios at roadside stations, I'm surprised to see that this is also the case at ambient stations. Is this because of the increase in ozone concentrations, and if so, does this match quantitatively with model results / stationary state estimates?

   The values given in Fig. 9 are also not in good agreement with the number of 0.7 given for the NO2/NOx ratio in section 2.3. Clearly, this ratio is not constant over the measurement period and varies strongly within the area. How will that impact on the results?

3. I do not see what I can learn from Fig. 7 which is not already shown in Fig. 6.

4. In section 3.2.1, a filtering of the data for congestion situations is described, and I can see the reason why the authors apply this filter. On the other hand, isn't there a risk of introducing a low bias, as the most busy (and thus most polluted)

parts of the roads which have the highest risk of congestion will be removed from the data?

5. If I understood the diurnal normalisation discussed in section 3.2.2 right, not the actual diurnal profile from the LP DOAS is used but rather the mean profile for that day of week, scaled to the actual LP DOAS measurements. As can be seen in Figure 5, the match is not very good between these two curves, and I'm wondering what that implies for the validity of the correction and the remaining bias from non-coincidence of measurements.

6. In Figure 8 and the discussion in the text, the measurements taken in March 2017 and December 2010 are used to characterise the long-term evolution of NO2 in Hong Kong. While the differences are large enough to be convincing, I still think that some discussion is needed here to exclude and quantify other effects such as weather, season or sampling.

7. In section 3.4, the differences between the magnitude of the NO2 concentrations measured by EPD ambient stations and on-road CE-DOAS is discussed in the context of Figure 11. However, already in Fig. 6 it can be seen that CE-DOAS values are on average clearly (much) higher than the station data, although measured within 100 m. I assume that this is mainly due to the different measurement altitudes and the steep vertical profile of NO2 in this urban environment (see also the earlier paper on the LP-DOAS measurements by Chan et al., 2012). In my opinion, this asks for some discussion with respect to the representativity of the CE-DOAS measurements and the station data, for example for human health and compliance with environmental legislation.

8. In order to put Figure 12 to use in other studies, it is important to know if this is a snapshot or an average over many observations. If the latter is true, the number of individual measurements that go into these averages and also the RMS are

relevant so that the reader can get an idea of how representative the mean value is.

9. I'm missing a statement on the availability of data – as the high resolution NO2 map is one of the main outcomes of the study, readers should know how to access it.

10. The text is overall well written and clear, but there are several shorter sections which need careful proof reading for grammar.
* * *

---

## Author Comment (AC1) · 5 Dec 2018

We thank reviewer #1 for the referee for the constructive comments. These comments are helpful for improving our manuscript. We understand that the comments are positive on the scientific content of the manuscript while appropriate revisions and clarifications are necessary. We have addressed the reviewer's comments on a point to point basis as below for consideration.

The manuscript 'Analysis of spatial and temporal patterns of on-road NO2 concentrations in Hong Kong' presents an investigation of the spatial and temporal variability of street level NO2 concentrations in Hong Kong. Two on road measurement campaigns were performed in 2010 and 2017 which combined both remote sensing LP-DOAS and

mobile in-situ CE-DOAS measurements. As the measurements were taken at different time under different conditions, the authors put a big effort on filtering and normalizing the data in order to make these data sets comparable. Details of the filtering and normalizing procedure are presented. The filtered data were used to analyze the long term trend as well as the temporal and spatial characteristic of NO2. The authors also show some interesting characteristics, e.g., enhancement of NO2 levels around shopping area during weekends and increase of NO2 around metro stations. The manuscript is in general well written and the topic is of interest of the general audiences of "Atmospheric Measurement Techniques". However, some clarifications are necessary. I suggest this manuscript for publication after a minor revision.

Specific comment:

As the measurement campaign were only performed in 2010 and 2017 each for a week. Although the authors have put a lot of effort in filtering and normalizing the data, it is difficult to say the measurements are representative for the general condition. I understood that it is difficult to have longer term measurements, but the authors should at least clarify that it is only a short term measurement and avoid using the term 'representative'. In addition, the measurement campaigns were performed in different seasons of the year and the meteorological conditions could be quite different. Is there any good reason?

Response: The measurement campaigns took place in different years and months and each for around one week time (10 days in Dec. 2010 and 8 days in Mar. 2017). It is very difficult to have regular measurements to derive annual average map. Therefore, we put a lot of effort on filtering and normalizing the data to get a better overview. In the revised manuscript, we rephased the term 'representative' to avoid confusion.

Regarding to the measurement campaign in different seasons, we tried to organize the campaigns in the similar time of year, but due to certain limitations, we can only measure in these two time frames. The two measurement campaigns were performed in

winter (December) and early spring (March). We have analyzed the meteorological parameters including temperature, humidity, wind speed and wind direction taken during the two measurement campaigns. The results show that the meteorological conditions are quite similar during the two campaigns. We have supplemented the information in the revised manuscript (page12, line 24-26).

Section 3.2.1: The authors present an example of the NO2 level under congestion condition and concluded that to filter data take 8 s after the vehicle speed below 5km/h. The lag time shown in this case is the combination of accumulation of NO2 in ambient plus the lag time of the instrument. The lag time of the instrument is fixed and can be measure, while the time of accumulation of NO2 varies. Clarification is necessary.

Response: A clarification is added to the revised manuscript (page8, line 1-2).

Figure 4 caption: Change 'during stops due to traffic congestion' to 'during traffic congestion'.

Response: The caption is revised according the reviewer's suggestion (page 8).

Section 3.3: Deriving long term trend from 2 weeks of measurements is not very convincing. The authors should state clear the purpose of comparing these short term measurements. The analysis of EPD and LP-DOAS data is variable though.

Response: The purposes of comparing these short term measurements are (1) to illustrate the differences between on-road mobile and road side stationary measurements and (2) to examine the consistency of the long term trend of road side NO2 derived from stationary measurements. We have supplemented the information in the revised manuscripts (page 9, line 16 to page 12, line 2).

Figure 6: This plot contains a lot of information already. However, I still would like to know whether it is possible to compare the EPD data measured at the same time when the CE-DOAS was passing by and how's the correlation in between? The labels in the plots should be larger. The date in the title of each plot is redundant, please remove

them.

Response: Our measurements only have few overpasses with the EPD stations during the campaigns. As a result, there are only few data points for comparison. Therefore, investigating the correlation for such a small dataset might not be statistical significance. We have revised Figure 6 according to reviewer comments (page 10).

Figure 8: How does the average map calculated for each year? Does it corrected for the diurnal effect? Since the authors described the measurements were taken during the different time of the day which contains the morning and evening rush hours and non-rush-hour at noon, it may lead to a bias in averaging all measurements.

Response: The measurements were taken with a fix schedule during morning rush hours, noontime and evening rush hours. We weighted the morning, noontime and evening measurements equally in the averaging process. Therefore, we do not correct for diurnal pattern of NO2. As both the measurements in 2010 and 2017 are processed with the same procedure, it is unlikely to have a bias when comparing the two datasets. We have supplemented a brief description of the averaging procedure in the figure caption (page 13).

Figure 11: NO2 concentration measured by the 7 EPD stations are shown on the map as well. But the authors don't describe any results such as the difference between EPD measurements and the coinciding closest on-road measurements, and the NO2 average changes in these 2 years of EPD stations.

Response: The EPD measurements shown on the maps are used to illustrate the consistency of the on road and stationary measurements. We have supplemented a brief discussion regarding the on road and stationary measurements (page 11, line 5 -9 and page 17, line 21). As we have discussed before, there are only few coinciding CE-DOAS and EPD measurement data. Looking into this small dataset might not be able to derive statistical significant conclusion.

Page 5 line 28: 'a equilibrium state' to 'an equilibrium state'.

Response: We have removed the sentence according to comment from reviewer #2.

Page 12 line 2-3: Explain why >1 ppb is significant. I suppose this is related to the detection limit of the instrument. Please specify it in the methodology section.

Response: The detection limit of the instrument is now included in the methodology section of the revised manuscript (page 4, line 7).

Page 15 line 10: 'primary NO is could be. . .' is grammatically incorrect. Please revise.

Response: The grammatically mistake has been corrected (page 15, line 26).

Page 17 line 3: If the traffic load is mostly constant in commercial areas which include most shopping malls on Sunday, why the differential map shows the decrease of NO2 around shopping malls? A better description is necessary.

Response: Although the traffic load is mostly constant, the parking spaces are limited in these shopping areas and results in low speed cruising and traffic congestion around these major shopping areas during weekend, which lead to higher emission of NOx. This explanation is written on page 19, line 7-10.

Page 18 line 14 and 15: Same as before, explain why >1 ppb is significant.

Response: The detection limit of the instrument is now included in the methodology section of the revised manuscript (page 4, line 7).

Page 20 line 20: I couldn't see the causal relationship between the increase amount of private cars and public transport usage with the significant increase of weekend drivers in Hong Kong. The authors should describe it better.

Response: Both number of private car and public transport usage increase in the past few years implied that the usage per private car is greatly reduced. The decrease of private car usage is mainly due to the reduction for daily commute using private cars,

which is coherent with the increase of public transport usage. As a result most of the private cars are mainly used during weekends. We have supplemented the explanation in the revised manuscript (page 19, line 3-5).

---

## Author Comment (AC2) · 5 Dec 2018

We thank reviewer #2 for careful reading our manuscript and the very detailed comments. They certainly helped us to improve the manuscript. We understand that the comments are positive on the scientific content of the manuscript while appropriate revisions and clarifications are necessary. We have addressed the reviewer's comments on a point to point basis as below for consideration.

In this manuscript, Zhu et al. report on measurements of boundary layer NO2 in Hong Kong using different techniques. In two campaigns, car-based measurements with a CE-DOAS instrument were performed for several days at different times of the day, covering both rush-hour and normal conditions. These measurements are complemented

by data from the in-situ measurement network, a long path DOAS instrument operating during and in between campaigns, and OMI satellite data. Data were analysed for their temporal trend, the diurnal profile, the week-end effect, their spatial distribution and the NO2 / NOx ratio.

The paper reports interesting measurement results from a highly polluted city enforcing strict emission controls and highlights some nice local effects such as changes in pollution levels around metro stations. The manuscript is overall well written but focuses on reporting measurement results and a qualitative interpretation. It therefore does not fit well into the scope of AMT ("The main subject areas comprise the development, intercomparison, and validation of measurement instruments and techniques of data processing and information retrieval for gases, aerosols, and clouds.") but should rather have been submitted to ACP in my opinion. It would also benefit from a more quantitative discussion including error bars.

Nevertheless, I recommend it for publication after the following points have been fully addressed.

Response: Before we submitted our manuscript to AMT, we carefully thought about the choice of journals and we chose AMT because our manuscript reports the application of mobile CE-DOAS, the data analysis method of the mobile measurements and the measurement results. We also compared our mobile measurements with in-situ monitor data. Therefore, we think the work represented fits well with the scope of AMT especially the specially issue of "Advances in cavity-based techniques for measurements of atmospheric aerosol and trace gases". We hope the manuscript is of interests for the general audience of the journal.

1. Was any correction applied to the in-situ chemiluminescence NOx analysers for cross-sensitivities?

Response: Cross-sensitivities correction is not applied to the chemiluminescence NOx analyzer measurements. However, the in-situ monitor operated by EPD have undergoes a series of calibration and verification procedures. The quality of the measurement data is proofed to meet the measurement standard.

2. I'm not yet convinced by the discussion of the NO2 to NOx ratios. While I can understand that the ratio is driven by the fraction of NOx emitted as NO close to the source, and therefore a change in technology used in the car fleet can have an impact on NO2 to NOx ratios at roadside stations, I'm surprised to see that this is also the case at ambient stations. Is this because of the increase in ozone concentrations, and if so, does this match quantitatively with model results/ stationary state estimates?

Response: The increase of ambient ozone certainly has an effect on the NO2 to NOx ratio. However, it is very difficult to quantify the contribution of increase of ozone on the increase of NO2 to NOx ratio even with chemical transport model. In addition, the focus of this section is to analysis the long term change of on-road and ambient NO2. Model study of the interaction between ambient O3 and NOx is certainly interesting but beyond of the scope of the paper. We have supplemented the information in the revised manuscript (page 16, line 3-5).

The values given in Fig. 9 are also not in good agreement with the number of 0.7 given for the NO2/NOx ratio in section 2.3. Clearly, this ratio is not constant over the measurement period and varies strongly within the area. How will that impact on the results?

Response: The NO2/NOx ratio given in section 2.3 reflects the general condition in Hong Kong and of course this number could vary in a wide range depending the local situation. The value indicated in section 2.3 is more representative for the ambient station measurements Fig 9b. The ratio provided is only supplementary information and not our focus of the study. In order to avoid confusion, we have removed the calculation of NO2/NOx ratio in section 2.3.

3. I do not see what I can learn from Fig. 7 which is not already shown in Fig. 6.

СЗ

Response: We would like to show the decrease of NO2 level is not only happening in Hong Kong, but also in the surrounding areas. Therefore, we provided the satellite images of NO2 spatial distribution over Pearl River Delta in both 2010 and 2017.

4. In section 3.2.1, a filtering of the data for congestion situations is described, and I can see the reason why the authors apply this filter. On the other hand, isn't there a risk of introducing a low bias, as the most busy (and thus most polluted) parts of the roads which have the highest risk of congestion will be removed from the data?

Response: The reviewer is right that the filter implemented would remove more data from busy roads where congestions happen more frequently. The filter is only applied to the maps for comparison (Figure 8), so that the maps are focus on NO2 concentrations instead of congestion patterns, since the congestion patterns could be very different due to road constructions and traffic accidents. The interpretation of the data is based on the difference between these maps, which they are analyzed in the same way. Therefore, the biases are very likely to be cancelled out with each other. On the other hand, this filter was not applied to the NO2 spatial distribution analysis (Figure 12), so it will not affect the interpretation of the spatial pattern of NO2. In order to avoid confusion, we have added a sentence in section 3.2.1 to clarify that the filter is only applied the maps for comparison (page 8, line 8).

5. If I understood the diurnal normalization discussed in section 3.2.2 right, not the actual diurnal profile from the LP DOAS is used but rather the mean profile for that day of week, scaled to the actual LP DOAS measurements. As can be seen in Figure 5, the match is not very good between these two curves, and I'm wondering what that implies for the validity of the correction and the remaining bias from non-coincidence of measurements.

Response: In this study, the mean profile of the day of week is scaled to fit the actual LP-DOAS measurement and the resulting profile is used for diurnal variation correction. Although the LP-DOAS measures along an optical path of 2km, the results may

still not be fully representative for the general condition of the entire Hong Kong. Figure 5 shows the LP-DOAS and 7 EPD monitor stations NO2 measurements. The result shows the scaled LP-DOAS data matches better the general condition of Hong Kong. Therefore, we use the scaled mean profile instead of the actual coinciding data. We have included a more detailed explanation of the use of scaled profiles instead of coinciding measurement in the revised manuscript (page 9, line 6-8).

6. In Figure 8 and the discussion in the text, the measurements taken in March 2017and December 2010 are used to characterise the long-term evolution of NO2 in Hong Kong. While the differences are large enough to be convincing, I still think that some discussion is needed here to exclude and quantify other effects such as weather, season or sampling.

Response: The measurement campaigns were held in different seasons, we tried to organize the campaigns in the similar time of year, but due to certain limitations, we can only measure in these two time frames. The two measurement campaigns were performed in winter (December) and early spring (March). We have analyzed the meteorological parameters including temperature, humidity, wind speed and wind direction taken during the two measurement campaigns. The results show that the meteorological conditions are quite similar during the two campaigns. We have supplemented the information in the revised manuscript (page 12, line 24-26).

7. In section 3.4, the differences between the magnitude of the NO2 concentrations measured by EPD ambient stations and on-road CE-DOAS is discussed in the context of Figure 11. However, already in Fig. 6 it can be seen that CE-DOAS values are on average clearly (much) higher than the station data, although measured within 100 m. I assume that this is mainly due to the different measurement altitudes and the steep vertical profile of NO2 in this urban environment (see also the earlier paper on the LP-DOAS measurements by Chan et al., 2012). In my opinion, this asks for some discussion with respect to the representativity of the CE-DOAS measurements and the station data, for example for human health and compliance with environmental

**legislation.**

Response: The vertical distribution of NO2 under on-road/road side conditions varies in a wide range. The EPD ambient stations are located at different altitude and in general above 10m a.g.l., while the road side stations are measuring at 3m a.g.l.. The inlet of our on-road mobile measurement is setup at 1.5m a.g.l, which is much closer to the pedestrians breathing height. As the tail pipes of vehicles are usually at 10-30cm a.g.l., our mobile measurement inlet is much closer to the emission sources and therefore in general measure higher NO2 concentrations. We have supplemented the description of the EPD stations (page 5, line 11-12) and the explanation of much higher on-road NO2 measured by the mobile CE-DOAS (page 12, line 5-9) in the revised manuscript.

8. In order to put Figure 12 to use in other studies, it is important to know if this is a snapshot or an average over many observations. If the latter is true, the number of individual measurements that go into these averages and also the RMS are relevant so that the reader can get an idea of how representative the mean value is.

Response: The spatial distribution of NO2 shown in Figure 12 is an average of all available measurements. The standard route measurement was performed 3 times per day, while other locations only have single or few overpasses during the two campaigns. Therefore, the resulting map can be regards as a consistent snapshot. In order to have a better overview, the data are corrected for diurnal variation using the LP-DOAS measurements (Figure 12b). To avoid confusion, we have extended the description of the figure capture of Figure 12 (page 20).

9. I'm missing a statement on the availability of data – as the high resolution NO2 map is one of the main outcomes of the study, readers should know how to access it.

Response: We have now included a statement on the availability of data. The mobile measurement data is available on request from the corresponding author (ka.chan@dlr.de).

10. The text is overall well written and clear, but there are several shorter sections which need careful proof reading for grammar. Response: We have carefully proofread the manuscript again and corrected the typing and grammatical errors.